# ZegOT: Zero-shot Segmentation Through Optimal Transport of Pixels to Text Prompts

## Abstract

Recent success of large-scale Contrastive Language-Image Pre-training (CLIP) has led to great promise in zero-shot semantic segmentation by transferring text-image aligned knowledge to pixel-level classification. However, existing methods usually require an additional image encoder or retraining/tuning the CLIP module. Here, we propose a novel **Z**ero-shot s**eg**mentation with **O**ptimal **T**ransport (ZegOT) method that matches multiple text prompts with frozen image embeddings through optimal transport. In particular, we introduce a novel Multiple Prompt Optimal Transport Solver (MPOT), which is designed to learn an optimal mapping between multiple text prompts and pixel embeddings of the frozen image encoder layers. This unique mapping method facilitates each of the multiple text prompts to effectively focus on distinct visual semantic attributes and diversify the learned knowledge to robustly cope with previously unseen categories. Through extensive experiments on benchmark datasets, we substantiate the effectiveness of our methods and demonstrate its superior performance in Zero-shot Semantic Segmentation (ZS3) settings.

## 1 Introduction

Zero-shot Semantic Segmentation (ZS3) is one of label-efficient approaches for dense prediction task, which reduces efforts for expensive pixel-level annotations of unseen object categories (Bucher et al., 2019). Vision language models (VLM) such as CLIP (Radford et al., 2021) have brought great advance in ZS3 task by transferring pre-trained text-image aligned knowledge to text-pixel level category matching problems (Ding et al., 2022; Xu et al., 2021; Zhou et al., 2022a; Rao et al., 2022; Zhou et al., 2022d). The key idea here is to use the VLM knowledge gained through contrastive learning on large-scale text-image pairs through a special knowledge transfer process tailored to ZS3.

Suppose that our goal is to learn a functional map that transfers the pre-trained image-level domain $\mathcal{X}$ knowledge to the novel pixel-level domain $\mathcal{Y}$. To achieve this goal, various approaches have been explored by training additional models or leveraging a novel domain knowledge, as indicated in Table 1. In general, existing ZS3 approaches based on pre-trained VLM can be categorized into two groups:

Table 1: Systemic analysis of ZegOT compared to CLIP-based segmentation models.

| Model | Proposal generator | CLIP module retrain/fine-tune | Multiple text prompts & pixel matching |
|---|---|---|---|
| ZegFormer Ding et al. (2022) | ✓ | ✗ | ✗ |
| zsseg Xu et al. (2021) | ✓ | ✗ | ✗ |
| MaskCLIP+ Zhou et al. (2022a) | ✗ | ✗ | ✗ |
| DenseCLIP Rao et al. (2022) | ✗ | ✓ | ✗ |
| ZegCLIP Zhou et al. (2022d) | ✗ | ✓ | ✗ |
| Freeseg Qin et al. (2023) | ✗ | ✓ | Cross-attention |
| MVP-SEG+ Guo et al. (2023) | ✗ | ✗ | OCL loss |
| **ZegOT (Ours)** | ✗ | ✗ | Optimal Transport |

*Note*: OCL; Orthogonal Constrained Loss

frozen image encoder with learnable proposal generator-based approaches (FE) (Ding et al., 2022; Xu et al., 2021), and trainable image encoder-based approaches (TE) (Rao et al., 2022). FE approaches utilize the proposal generator to gain domain $\mathcal{Y}$ knowledge from the image-level classification capability. A major drawback of FE approaches is that performance is highly dependent on the frozen VLM. On the other hand, TE approaches explicitly address the VLM dependency issue by retraining or fine-tuning the image encoder part. However, since the image encoder is optimized for a specific domain, network parameters can be overfitted to seen distribution, resulting sub-optimal segmentation performance for unseen distribution.

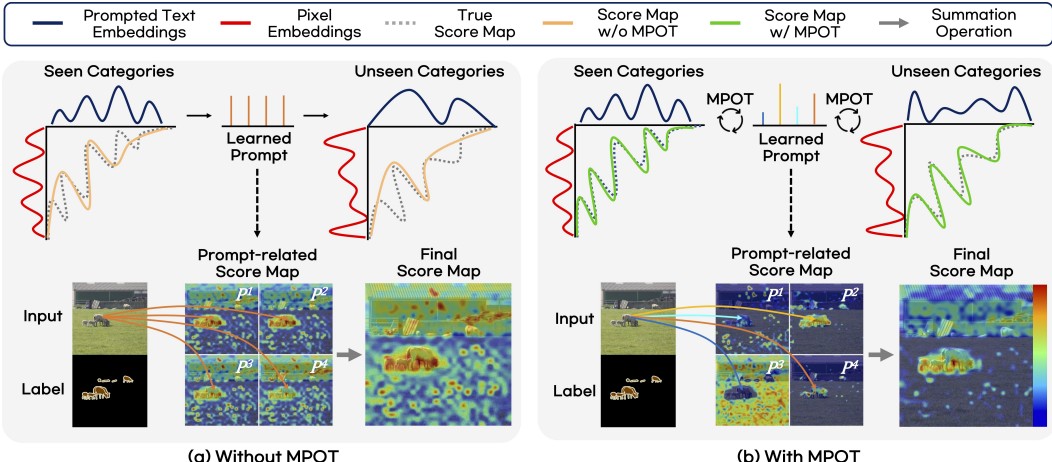

Figure 1: Visualization of text-pixel alignment. (a) Without proposed MPOT, all the learned multiple prompts $P^i$ are cohered and their related score maps resemble each others, which leads to a mismatch with the true score map. (b) On the other hand, with our MPOT, each $P^i$ is diversely projected and their related score map focuses on different semantic attributes, which results in robustness for handling the unseen categories. As a result, the final score map with MPOT effectively focuses on the ground truth regions compared to that of without MPOT.

One naive solution to mitigate the overfitting issue is to keep the entire VLM module frozen, and to introduce small amount of learnable parameters such as text prompts, which can be borrowed from the recent success of prompt learning for fine-tuning language models (Petroni et al., 2019; Shin et al., 2020; Jiang et al., 2020; Liu et al., 2021). These light-weight text prompts learn the desired downstream task knowledge and transfer it to unseen distribution, while mitigating the issue of catastrophic forgetting of pre-trained VLM knowledge. However, this solution is still problematic because the introduced text prompts and pixel embeddings from the frozen VLM are passively aligned, which leads the learned prompts to be converged to represent similar semantic features as shown in Figure 1(a). Additionally, this solution lacks diversity in learned prompts to robustly handle the unseen categories, thus fails to match the true score map of the text and pixel embeddings.

To further boost the effectiveness of learning the multiple text prompts, here we introduce optimal transport (OT)-based multi-prompt tuning method, by expanding the concept addressed in (Chen et al., 2022). Our motivation for utilizing OT is originated from its luminous property of distribution matching without the need for additional trainable parameters, which enables the multiple text prompt distribution to be optimally matched with the desired pixel distribution. As provided in Figure 1(b), we observe that our proposed multi-prompt optimal transport (MPOT) module, incorporating the concept of OT, effectively addresses the aforementioned problem by diversifying the learned prompt distribution without additional learnable parameters. With our MPOT, each text prompt-related score map selectively focuses on specific semantic features. Consequently, the refined text prompts provide diversity that robustly cope with the unseen categories, which leads to facilitate the optimal alignment of the text and pixel embeddings. These individual maps are then combined to yield the final score map, which efficiently attends to the ground truth regions. Due to the lack of the additional trainable parameters, our method is less prone to overfitting issue, resulting in the state-of-the-art (SOTA) performance on ZS3 tasks. Our contributions can be summarized as:

- We propose a novel framework, called ZegOT, which introduces a multi-prompt optimal transport module that effectively aligns the distribution between text and pixel embeddings from the frozen VLM encoders.

- We explore the role of the optimally aligned text-pixel score maps as contributing intermediate features for zero-shot semantic segmentation task, avoiding overfitting on seen classes.

- Through extensive experiments on three benchmark datasets, we demonstrate that our ZegOT achieves the superior performance for zero-shot semantic segmentation tasks compared to the previous approaches.

## 2 RELATED WORK

### 2.1 PROMPT LEARNING

Prompt learning has been initially introduced in the field of Natural Language Processing (NLP), which efficiently adapts large-scale model knowledge to various downstream tasks. Rather than using traditional fine-tune methods to transfer the large-scale model knowledge to downstream tasks, the prompt learning formulates the downstream adjustment problem by training light-weight optimal textual prompts (Petroni et al., 2019; Shin et al., 2020; Jiang et al., 2020; Liu et al., 2021). Compared to the fine-tuning method, text prompts-driven downstream adaptation is efficient, but still alleviates the domain shift problem that occurs between the pre-trained domain and the downstream domain. Recently, introducing a set of learnable text prompts into the frozen VLM achieved superior performance in various computer vision (CV) tasks (Zhou et al., 2022c;b; Gao et al., 2021; Rao et al., 2022). Visual Prompt Tuning (VPT) is also a novel solution that introduces trainable image prompts to each layer of a transformer encoder (Jia et al., 2022), which can be transferred to various downstream tasks (Zang et al., 2022; Sohn et al., 2022; Liu et al., 2022). Our method utilizes multiple text prompts to adapt VLM to the segmentation task, while keeping the entire encoder part frozen.

### 2.2 ZERO-SHOT & OPEN-VOCABULARY SEMANTIC SEGMENTATION

Semantic segmentation is a core computer vision task to densely analyze visual context. Recent success of CLIP (Radford et al., 2021) accelerates the advancement of language-driven semantic segmentation by utilizing pre-trained knowledge of VLM (Li et al., 2022; Xu et al., 2022; Liang et al., 2022). However, since this dense prediction task requires a labor-intensive pixel-level annotation, there arise a label-imbalance issue, *i.e.,* not all the categories are annotated in the training dataset. Zero-shot semantic segmentation (ZS3) solves this label-imbalance problem by generalizing labeled (seen) class knowledge to predict new (unseen) class information (Bucher et al., 2019), which can be further expanded to open-vocabulary segmentation, which aims to segment totally unseen categories during the training procedure. MaskCLIP+ (Zhou et al., 2022d) introduces a ZS3 method by simply extracting the text-driven visual features from the CLIP image encoder. ZegCLIP (Zhou et al., 2022d) successfully bridges the performance gap between the seen and unseen classes by adapting a visual prompt tuning technique instead of fine-tuning the frozen CLIP image encoder. Recently, FreeSeg (Qin et al., 2023) and MVP-SEG+ (Guo et al., 2023) introduce text prompt-driven method for realizing open-vocabulary segmentation. In specific, MVP-SEG+ employs Orthogonal Constraint Loss (OCL) to each prompt to exploit CLIP features on different object parts. On the other hand, our proposed ZegOT tries to utilize CLIP knowledge by introducing a novel Multiple Prompt Optimal Transport Solver (MPOT), which matches distribution between text prompts and pixels.

ZS3 can be performed by either inductive or transductive settings. Compared to inductive ZS3 where class names and pixel-level annotations of unseen classes are both unavailable during training (Ding et al., 2022), a newly introduced transductive setting boosts the ZS3 performance by utilizing unseen class names and self-generated pseudo labels guided by the model itself during training (Gu et al., 2020; Xu et al., 2021; Pastore et al., 2021; Zhou et al., 2022a;d). We adapt the transductive setting, which can achieve comparable performance to that of the supervised learning method.

### 2.3 OPTIMAL TRANSPORT

Optimal transport (OT) is a general mathematical framework to evaluate correspondences between two distributions. Thanks to the luminous property of distribution matching, the optimal transport has received great attention and proven its generalization capability in various computer vision tasks, such as domain adaptation (Flamary et al., 2016), semantic correspondence problem (Liu et al., 2020), graph matching (Xu et al., 2019a;b), and cross-domain alignment (Chen et al., 2020), etc. Among various methods, Sinkhorn algorithm can efficiently solve the OT problem through entropy-regularization (Cuturi, 2013), and it can be directly applied to deep learning frameworks thanks to the extension of Envelop Theorem (Peyré et al., 2019). Prompt Learning with Optimal Transport (PLOT) (Chen et al., 2022) is mostly related to ours, which optimizes the optimal transport distance to align visual features and text features by the Sinkhorn algorithm given trainable multiple text prompts. However, PLOT exclusively focuses on few-shot image-level prediction tasks, while we apply the optimal transport theory to zero-shot dense prediction problems, which facilitates ZS3.

## 3 PRELIMINARY

**Optimal Transport Problem**  Optimal transport aims to minimize the transport distance between two probability distributions. In this paper, we only consider discrete distribution which is closely related to our framework. We assume discrete empirical distributions $\boldsymbol{\mu}$ and $\boldsymbol{\nu}$ that are defined on probability space $\mathcal{F}, \mathcal{G} \in \Omega$, respectively, as follows:

$$\boldsymbol{\mu} = \sum_{i=1}^{M} p_i \delta_{f_i}, \quad \boldsymbol{\nu} = \sum_{j=1}^{N} q_j \delta_{g_i}, \tag{1}$$

where $\delta_f$ and $\delta_g$ denote Dirac functions centered on $f$ and $g$, respectively, $M$ and $N$ denote the dimension of the empirical distribution. The weight vectors $\boldsymbol{p} = \{p_i\}_{i=1}^{M}$ and $\boldsymbol{q} = \{q_i\}_{j=1}^{N}$ belong to the $M$ and $N$-dimensional simplex, respectively, $i.e.,$ $\sum_{i=1}^{M} p_i = 1$, $\sum_{j=1}^{N} q_j = 1$. The discrete optimal transport problem can be then formulated as:

$$\boldsymbol{T}^* = \underset{\boldsymbol{T} \in \mathbb{R}^{MXN}}{\arg\min} \sum_{i=1}^{M} \sum_{j=1}^{N} \boldsymbol{T}_{ij} \boldsymbol{C}_{ij}$$
$$\text{s.t.} \quad \boldsymbol{T}\boldsymbol{1}^N = \boldsymbol{\mu}, \quad \boldsymbol{T}^\top \boldsymbol{1}^M = \boldsymbol{\nu}. \tag{2}$$

Here, $\boldsymbol{T}^*$ is called the optimal transport plan, which is learned to minimize the total distance between the two probability vectors. $\boldsymbol{C}$ is the cost matrix which represents the distance between $\boldsymbol{f}_i$ and $\boldsymbol{g}_j$, $e.g.,$ the cosine distance $\boldsymbol{C}_{ij} = 1 - \frac{\boldsymbol{f}_i \boldsymbol{g}_j^\top}{\|\boldsymbol{f}_i\|_2 \|\boldsymbol{g}_j\|_2}$, and $\boldsymbol{1}^M$ refers to the $M$-dimensional vector with ones.

However, solving the problem (2) costs $O(n^3 \log n)$-complexity ($n$ proportional to $M$ and $N$), which is time-consuming. This issue can be efficiently solved by entropy-regularizing the objective, called the Sinkhorn-Knopp (or simply Sinkhorn) algorithm (Cuturi, 2013). In Sinkhorn algorithm, the optimization problem is reformulated as:

$$\boldsymbol{T}^* = \underset{\boldsymbol{T} \in \mathbb{R}^{MXN}}{\arg\min} \sum_{i=1}^{M} \sum_{j=1}^{N} \boldsymbol{T}_{ij} \boldsymbol{C}_{ij} - \lambda H(\boldsymbol{T})$$
$$\text{s.t.} \quad \boldsymbol{T}\boldsymbol{1}^N = \boldsymbol{\mu}, \quad \boldsymbol{T}^\top \boldsymbol{1}^M = \boldsymbol{\nu}. \tag{3}$$

where $H(\boldsymbol{T}) = \sum_{ij} \boldsymbol{T}_{ij} \log \boldsymbol{T}_{ij}$ and $\lambda > 0$ is the regularization parameter. For the problem (3), we have an optimization solution with fewer iterations as follows:

$$\boldsymbol{T}^* = \text{diag}(\boldsymbol{a}^t) \exp(-\mathbf{C}/\lambda) \text{diag}(\boldsymbol{b}^t) \tag{4}$$

where $t$ is the iteration and $\boldsymbol{a}^t = \boldsymbol{\mu}/\exp(-\mathbf{C}/\lambda)\boldsymbol{b}^{t-1}$ and $\boldsymbol{b}^t = \boldsymbol{\nu}/\exp(-\mathbf{C}/\lambda)\boldsymbol{a}^t$, with the initialization on $\boldsymbol{b}^0 = \mathbf{1}$.

## 4 METHODS

In this section, we present a method for performing zero-shot segmentation tasks using our proposed ZegOT framework as illustrated in Figure 2. A primary goal of ZegOT is to segment objects belong to both seen classes $\mathcal{C}_S$ and unseen classes $\mathcal{C}_U$, $i.e.,$ $\mathcal{C} = \mathcal{C}_S \cup \mathcal{C}_U$, where $\mathcal{C}_S \cap \mathcal{C}_U = \emptyset$. Note that class names of $\mathcal{C}_U$ are further provided during training under the transductive ZS3 setting. To clarify, we define several notations within our ZegOT framework: a pair of frozen CLIP text encoder $E_{\text{txt}}$ and image encoder $E_{\text{img}}$, trainable multiple text prompts $\mathcal{P}$, the global text alignment (GA) module, and trainable decoder $D_\theta$. Furthermore, we propose three fundamental components of ZegOT: (a) deep pixel alignment (DFA), (b) multi-prompt optimal transport (MPOT), and (c) inference with ensemble. In Sec. 4.1, we introduce the concept of learnable prompts-guided text embeddings and pixel embeddings from the frozen CLIP encoder and explain how they operate within our framework. In Sec. 4.2, we provide a detailed account of our ZegOT, including aforementioned three key components. Lastly, in Sec. 4.3, we describe the training procedure for our method and introduce the associated loss functions.

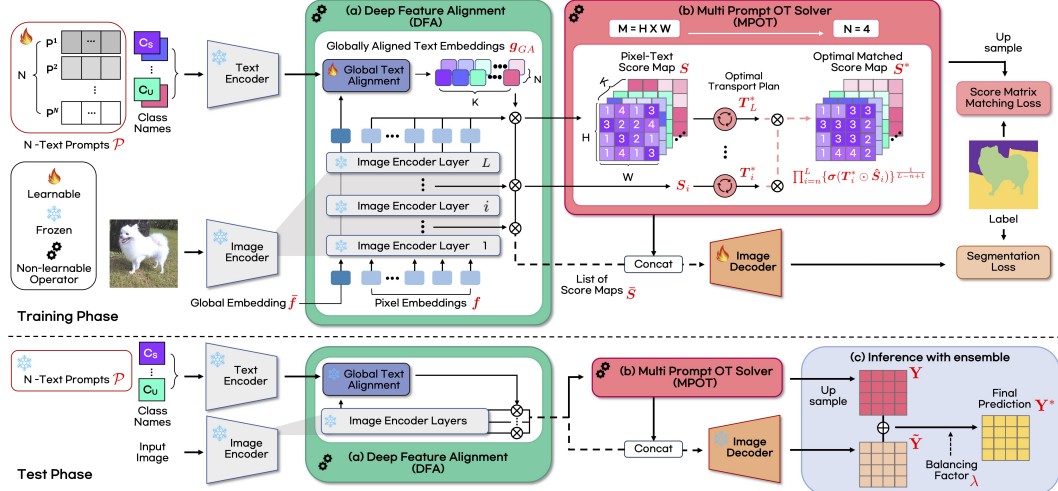

Figure 2: Overview of our proposed ZegOT for zero-shot semantic segmentation. (a) DFA: we deeply align all the intermediate pixel embeddings from the frozen image encoder with the prompted text embedding. (b) MPOT: given the text-pixel score maps from DFA, we adapt optimal transport plans for each score map and merge them using the geometric mean to obtain the final score map. (c) Inference with ensemble: ZegOT ensembles both the predictions from MPOT and the decoder with a balancing factor $\lambda$.

## 4.1 LEARNABLE PROMPTS-GUIDED TEXT-PIXEL EMBEDDING

To transfer CLIP's pre-trained knowledge, we create a set of $N$ text prompts denoted as $\mathcal{P} = \{\boldsymbol{P}^i|_{i=1}^N\}$ to effectively fine-tune the frozen CLIP encoder. Each text prompt $\boldsymbol{P}^i$ can be defined as $\boldsymbol{P}^i = [P_1^i, P_2^i, \cdots, P_l^i]$, where $l$ represents the length of the context tokens. These randomly initialized text prompts are then consistently added in front of $K$ tokenized class names, forming a set denoted as $\mathcal{T} = \{\{\mathcal{P}, \boldsymbol{c}^k\}|_{k=1}^K\}$. Note that the same text prompts $\mathcal{P}$ is shared across all class names. Here, $\{\boldsymbol{c}^k|_{k=1}^K\}$ represents the word embeddings of each class name, drawn from a larger set $\mathcal{C}$. Then, the set $\mathcal{T}$ is inputted to the frozen CLIP text encoder, resulting in the text embeddings $\boldsymbol{g} \in \mathbb{R}^{KN \times D}$, where $D$ represents the embedding dimension.

Now, when we input an image, it goes through the frozen CLIP image encoder, resulting in the global image embedding $\bar{\boldsymbol{f}}_{\boldsymbol{L}} \in \mathbb{R}^{1 \times D}$ and the $i$-th intermediate pixel embedding $\boldsymbol{f}_i \in \mathbb{R}^{H_L W_L \times D}$ for $i = 1, \cdots, L$. Here, $H_L$ and $W_L$ correspond to height and width of the pixel embeddings from the $L$-th layer. To merge these CLIP's text and pixel embeddings, we utilize the global text alignment (GA) module following an approach introduced in (Zhou et al., 2022d) to prepare $\boldsymbol{g}_{GA}$ to transfer the pretrained CLIP knowledge as follows:

$$\boldsymbol{g}_{GA} = \mathcal{Q}(\mathrm{cat}\left[\bar{\boldsymbol{f}}_{\boldsymbol{L}} \odot \boldsymbol{g}, \boldsymbol{g}\right]) \in \mathbb{R}^{KN \times D}$$

where $\mathcal{Q}$ denotes a linear layer for matching the concatenated embedding dimension to the original dimension $D$, and cat is the concatenation operator, $\odot$ is the Hadamard product.

## 4.2 ZERO-SHOT SEGMENTATION THROUGH OPTIMAL TRANSPORT (ZEGOT)

**(a) Deep Feature Alignment (DFA)** Using $\boldsymbol{g}_{GA}$, we leverages intermediate $i$-th pixel embeddings from image encoder layers to yield the desirable text-pixel aligned score map $\boldsymbol{S}_i$ as follows:

$$\boldsymbol{S}_i = \boldsymbol{f}_i \boldsymbol{g}_{GA}^\top, \tag{5}$$

where the superscript $^\top$ refers to the transpose operation, and $\boldsymbol{f}_i$ and $\boldsymbol{g}_{GA}$ are $\mathcal{L}_2$ normalized along the embedding dimension. The score map $\boldsymbol{S}_i \in \mathbb{R}^{H_L W_L \times KN}$ can serve as an input for the decoder or undergoes further refinement via the optimal transport plan, as depicted in Figure 2(b). Instead of relying on a single score map from the last layer, we utilize all the intermediate score maps to

fully leverage the frozen CLIP's knowledge. We refer to this multi-layer alignment as Deep Feature Alignment (DFA), which is defined as:

$$\bar{S} = \{\mathcal{M}(S_i)|_{i=1}^L\}. \tag{6}$$

where $\bar{S}$ is a list of score maps through DFA, $\mathcal{M}: \mathbb{R}^{H_L W_L \times KN} \rightarrow \mathbb{R}^{H_L W_L \times K}$ is the operation which first reshapes $\mathbb{R}^{H_L W_L \times KN} \rightarrow \mathbb{R}^{H_L W_L \times K \times N}$ and performs summation of all the score maps along the $N$ dimension. Details of DFA is further explained in Supplementary Material G.

**(b) Multi Prompt Optimal Transport Solver (MPOT)**   To transport the distribution of the multiple text prompts to pixel distribution, we first define the total cost matrix $C$ in (3) using the text-pixel aligned score map $S_i$ in (5). Specifically, for the $i$-th layer, we set $C := C_i = 1 - S_i$, where $C_i \in \mathbb{R}^{H_L W_L \times KN}$ denotes the $i$-th cost matrix. Given the cost matrix $C_i$, the goal of MPOT is to obtain the corresponding optimal transport plan $T_i^* \in \{T_i^*\}_{i=1}^L$ as given in (4). The role of the optimal transport plan is then to assign each of the $M = H_L W_L$ pixels to the $N$ prompts, allowing for multiple prompts to be associated with each pixel. This optimal transport plan represents a mapping matrix that maximizes the cosine similarity between the frozen pixel embedding and the text embeddings derived from learnable multiple prompts, as outlined in Algorithm 1 in Supplementary material A. Therefore, we reformulate the refined intermediate score map as:

$$S_i^* = T_i^* \odot S_i \tag{7}$$

where $S_i^*$ is the refined score map from $i$-th pixel embedding by adapting the transport plan $T_i^*$. Similar to DFA, instead of optimizing the transport plan just for a single layer, we leverages intermediate pixel-text score maps to yield multi-layer transport plans $\{T_i^*\}_{i=n}^L$, where $T_i^* \in \mathbb{R}^{H_L W_L \times KN}$ and $n$ denotes the starting layer that comprises these multi-layer transport plans. When combining these refined score maps, we apply a geometric mean, as follows:

$$S^* = \prod_{i=n}^L \sigma\{S_i^*\}^{\frac{1}{d}}, \tag{8}$$

where $d = L - n + 1$ denotes the depth of incorporating image encoder layers, $\sigma$ is the sigmoid function if $n \leq i < L$, and the identity function otherwise. It's worth noting that we apply the sigmoid function to all the refined intermediate score maps except for the last layer because the intensity range of all the score maps is (-1,1). Note that the transport plan $T^*$ in (4) only contains matrix multiplication and exponential operation, thus MPOT solver is fully differentiable and the gradients can be back-propagated throughout the entire neural network. Further analysis of efficiency and normalization of MPOT will be discussed in Supplementary material J and K.

**(c) Inference with ensemble**   The refined score map $S^*$ in (8) can be either directly utilized as a stand-alone logit or appended to decoder as an intermediate feature. Firstly, $S^*$ is reshaped and upsampled to be a stand-alone logit, as follows:

$$\mathbf{Y} = \mathcal{U}(\mathcal{M}(S^*)). \tag{9}$$

where $\mathbf{Y} \in \mathbb{R}^{HW \times K}$ is the prediction of the refined score map, and $\mathcal{U}: \mathbb{R}^{H_L W_L \times K} \rightarrow \mathbb{R}^{HW \times K}$ is the upsampling operator ($H_L W_L < HW$), where $H$ and $W$ are height and width of the input image, respectively.

Rather than soley reling on the prediction (9), we further utilize the learnable decoder to boost the segmentation performance. The score maps obtained through DFA in (6) can be incorporated as input for the decoder along with the refined score map $S^*$ as follows:

$$\tilde{\mathbf{Y}} = \mathcal{D}_\theta(\text{cat}[\bar{S}, \mathcal{M}(S^*)]). \tag{10}$$

where $\tilde{\mathbf{Y}} \in \mathbb{R}^{HW \times K}$ is the output of decoder. In order to synergistically exploit the collective knowledge derived from both the refined score map $\mathbf{Y}$ and more generalized information encapsulated in $\tilde{\mathbf{Y}}$ and, we formulate the final segmentation output $\mathbf{Y}^*$ as follows:

$$\mathbf{Y}^* = \lambda \cdot \mathbf{Y} + (1 - \lambda) \cdot \tilde{\mathbf{Y}} \tag{11}$$

where $\lambda \in [0, 1]$ denotes the hyper-parameter for controlling balance between the score map $\mathbf{Y}$ and image decoder outputs $\tilde{\mathbf{Y}}$. The balance factor $\lambda$ is set to 0.8 through component analysis in Supplementary Material H.

## 4.3 TRAINING PROCEDURE

In the training procedure, we follow the previous SOTA methods (Zhou et al., 2022d;a), where the entire training schedule is divided into two phases: seen classes-guided learning, and self-training. The details of the training phases are deferred in Supplementary material B.

**Loss Function** In this work, we combine three different losses which is similar to previous methods as follows:

$$\mathcal{L}_{\text{seg}} = \lambda_{\text{ce}}\mathcal{L}_{\text{ce}} + \lambda_{\text{fc}}\mathcal{L}_{\text{fc}} + \lambda_{\text{dc}}\mathcal{L}_{\text{dc}}, \quad \mathcal{L}_{\text{total}}(\Theta, \theta) = \mathcal{L}_{\text{seg}}(\mathbf{Y}, \mathbf{Y}^{\text{gt}}; \Theta) + \mathcal{L}_{\text{seg}}(\tilde{\mathbf{Y}}, \mathbf{Y}^{\text{gt}}; \Theta, \theta) \quad (12)$$

where $\Theta = [\mathcal{P}, \mathcal{Q}]$ which contains learnable multiple text prompts $\mathcal{P}$ and the linear layer $\mathcal{Q}$, $\mathcal{L}_{\text{seg}}$ denotes the segmentation loss combining different three losses. $\mathcal{L}_{\text{ce}}$, $\mathcal{L}_{\text{fc}}$ and $\mathcal{L}_{\text{dc}}$ are the cross entropy loss, the focal loss, and the dice loss, with $\lambda_{\text{ce}}$, $\lambda_{\text{fc}}$, and $\lambda_{\text{dc}}$ as corresponding hyper-parameters, respectively. $\mathbf{Y} \in \mathbb{R}^{HW \times K}$ and $\tilde{\mathbf{Y}} \in \mathbb{R}^{HW \times K}$ are the predictions of our model which is defined by (9) and (10), respectively. $\mathbf{Y}^{\text{gt}} \in \mathbb{R}^{HW \times K}$ is the ground-truth label. The details of the loss function are described in Supplementary material C.

## 5 EXPERIMENTS

### 5.1 DATASET AND EVALUATION METRIC

**Dataset** To evaluate the effectiveness of our proposed method, we carry out extensive experiments on three challenging datasets: PASCAL VOC 2012 (PAS20) (Everingham & Winn, 2012), PASCAL Context (PC-59) (Mottaghi et al., 2014) , and COCO-Stuff164K (COCO-170) (Caesar et al., 2018). For fair comparison with previous methods (Bucher et al., 2019; Xu et al., 2021; Ding et al., 2022; Zhou et al., 2022a;d), we follow the identical protocol of dividing seen and unseen classes for each dataset. The dataset details are described in Supplementary material D.

**Evaluation Metric** We measure the mean of class-wise intersection over union (mIoU) on both seen and unseen classes, indicated as mIoU(S) and mIoU(U), respectively. We adopt the harmonic mean IoU (hIoU) of seen and unseen classes as a primary metric. More details are deferred to Supplementary material E.

### 5.2 IMPLEMENTATION DETAILS

We implement the proposed method on the open-source toolbox MMSegmentation (Contributors, 2020) [1]. We conduct our algorithm on at most 8 NVIDIA A100 GPUs with batch size of 16. We adopt the pre-trained CLIP ViT-B/16 model[2] as the frozen encoder module for all the experiments. We set the number of multiple text prompts as $N = 4$, context length to 8, and MPOT layers $L = 4$. For the image decoder, we adopt FPN which is equipped with atrous spatial pyramid pooling (ASPP) (Chen et al., 2017) module. Further details of implementation, the image decoder, and the component analyses of related hyper-parameter are deferred to Supplementary material F, G, and H, respectively.

### 5.3 EXPERIMENTAL RESULTS

**Zero-shot Semantic Segmentation** Quantitative zero-shot segmentation results are presented in Table 2. Our ZegOT achieves the SOTA performance for PAS20 and PC-59, and showed the second-best performance for COCO-170. Figure 3 shows the qualitative zero-shot segmentation performance of our ZegOT and other previous approaches. Our method provides segmentation maps for unseen class objects most accurately compared to the previous SOTA methods (see the red arrows). More visual results are provided in Supplementary material N. To further evaluate the generalization capabilities of ZegOT, we performed a cross-domain experiment between COCO-170 and PC-59, as shown in Table 3. Our method demonstrates the great capability of generalization performance compared to the previous SOTA method in both experimental setting.

---

[1] https://github.com/open-mmlab/mmsegmentation
[2] https://github.com/openai/CLIP

Table 2: Quantitative comparison of zero-shot semantic segmentation performance of ZegOT with baseline methods on PAS20, PC-59, and COCO-Stuff 164K datasets. The **bold** indicates the best performance and the underline indicates the second best performance, respectively.

| Methods | PASCAL VOC 2012 (PAS20) | | | PASCAL Context (PC-59) | | | COCO-Stuff164K (COCO-170) | | |
|---|---|---|---|---|---|---|---|---|---|
| | mIoU (U) | mIoU (S) | hIoU | mIoU (U) | mIoU (S) | hIoU | mIoU (U) | mIoU (S) | hIoU |
| CaGNet (Gu et al., 2020) | 30.3 | 78.6 | 43.7 | 16.3 | 36.4 | 33.5 | 13.4 | 35.6 | 19.5 |
| SPNet (Xian et al., 2019) | 25.8 | 77.8 | 38.8 | - | - | - | 26.9 | 34.6 | 30.3 |
| STRICT (Pastore et al., 2021) | 35.6 | 82.7 | 49.8 | - | - | - | 30.3 | 35.3 | 32.6 |
| zsseg (Xu et al., 2021) | 78.1 | 79.2 | 79.3 | - | - | - | 43.6 | 38.1 | 41.5 |
| MaskCLIP+ (Zhou et al., 2022a) | 88.1 | 86.1 | 87.4 | 66.7 | 48.1 | 53.3 | 54.7 | 39.6 | 45.0 |
| ZegCLIP (Zhou et al., 2022d) | 89.9 | **92.3** | 91.1 | 68.5 | 46.8 | 55.6 | **59.9** | 40.7 | **48.5** |
| FreeSeg(Qin et al., 2023) | 82.6 | 91.8 | 86.9 | - | - | - | 49.1 | **42.2** | 45.3 |
| MVP-SEG+(Guo et al., 2023) | 87.4 | 89.0 | 88.0 | 67.5 | 48.7 | 54.0 | 55.8 | 39.9 | 45.5 |
| ZegOT (Ours) | **90.9** | 91.9 | **91.4** | **72.5** | **50.5** | **59.5** | 57.5 | 38.7 | 46.2 |

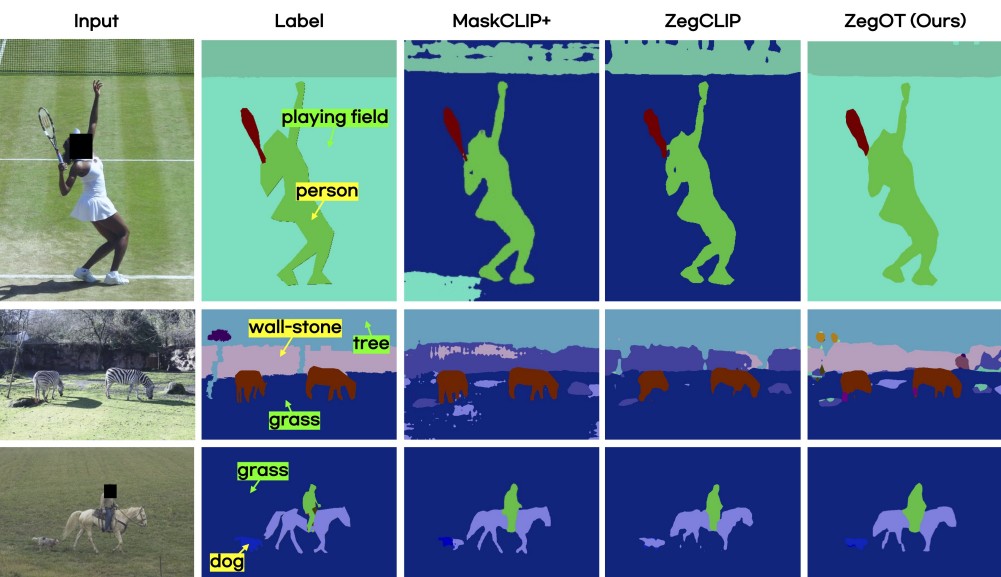

Figure 3: Qualitative segmentation performance comparison with the previous SOTA models. The green tag indicates unseen classes, while the yellow tag indicates seen classes.

**MPOT-driven Text Prompt-Image Alignment** In Figure 1, we validate the effectiveness of MPOT by visualizing each text prompt-related score map. We analyze the contribution of MPOT to produce better segmentation results for both seen and unseen categories, as further displayed in Figure 6. To further explore the origin of superior zero-shot segmentation performance of our ZegOT, we further visualize the distribution of the learned text embeddings using t-SNE projection (Hinton & Roweis, 2002) in Figure 4. We plot the average of prompted text embeddings for each class from the PAS20, which are represented by different colors. With MPOT, we observe that prompted text embeddings for every class are diversely dispersed, whereas the baseline method shows a significantly concentrated distribution. The result suggests that MPOT is helpful in distinguishing each class's attributes for both seen and unseen categories, leading to performance improvements in prompted class name-guided semantic segmentation.

## 5.4 ABLATION STUDIES

**Learnable Module** To demonstrate the effectiveness of our ZegOT framework, we conduct experiments by varing learnable modules within our framework as in Table 4(a). We firstly replace the learnable text prompt $\mathcal{P}$ with hand-crafted text prompts. Compared to our model, using hand-crafted prompts achieves inferior performance with large margins for both the unseen and seen classes. We further alter each of frozen CLIP encoder parts to be learnable and find that fully fine-tuning the model encoder parts resulted in a drastic segmentation performance drop on unseen classes,

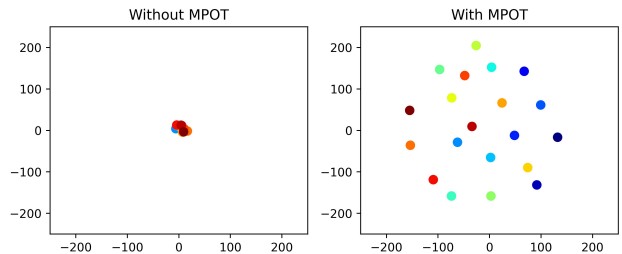

Figure 4: Comparison of t-SNE projections of learned text embedding with or without MPOT.

Table 3: Generalization performance on cross-domain datasets.

| Method | Source | Target |
|---|---|---|
| | | PC-59 |
| ZegFormer | COCO-170 | 36.1 |
| ZegCLIP | (Seen) | 41.2 |
| Ours | | **41.3** |
| Method | Source | COCO-170 |
| ZegCLIP | PC-59 | 16.7 |
| Ours | (Seen) | **17.8** |

Table 4: Ablation studies for the main proposed component.

| | Learnable module | Matching method | PASCAL VOC 2012 (PAS20) | | | PASCAL Context (PC-59) | | |
|---|---|---|---|---|---|---|---|---|
| | | | mIoU (U) | mIoU (S) | hIoU | mIoU (U) | mIoU (S) | hIoU |
| Ours | $\mathcal{D}_\theta, \mathcal{P}$ | MPOT | **90.9** | 91.9 | **91.4** | **72.5** | 50.5 | 59.5 |
| (a) | $\mathcal{D}_\theta$, (Hand-crafted $\mathcal{P}$) | MPOT | 78.1 | 83.7 | 80.8 | 54.9 | 33.5 | 41.6 |
| | $\mathcal{D}_\theta, \mathcal{P}, E_{\text{img}}$ | | 23.1 | **93.8** | 37.1 | 66.0 | **56.9** | **61.1** |
| | $\mathcal{D}_\theta, \mathcal{P}, E_{\text{img}}, E_{\text{txt}}$ | | 17.7 | 92.8 | 29.7 | 65.4 | 56.7 | 60.7 |
| (b) | $\mathcal{D}_\theta, \mathcal{P}$ | ✗ | 85.4 | 91.9 | 88.5 | 65.0 | 51.0 | 57.1 |
| | | Self-attention | 84.4 | 91.4 | 87.7 | 59.3 | 50.6 | 54.6 |
| | | Bipartite | 82.3 | 89.0 | 85.7 | 57.6 | 46.7 | 51.6 |

*Note*: $\mathcal{D}_\theta$; Image decoder, $\mathcal{P}$; Text prompts, $E_{\text{img}}$; CLIP image encoder, $E_{\text{txt}}$; CLIP text encoder

while it produced superior performance for seen classes. This could be attributed to the potential overfitting issue during fine-tuning. The experimental results support the idea that our superior zero-shot segmentation performance originates from the incorporation of the frozen CLIP encoders with only a few learnable modules, thereby avoiding overfitting on seen classes. Further ablation studies on learnable parameters are provided in Supplementray material I.

**Matching Method**  In Table 4(b), we further ablate our proposed text-pixel matching method. Our model without MPOT shows inferior performance on unseen classes with margins of above 5% mIoU, which supports that MPOT efficiently helps the learnable prompts $\mathcal{P}$ to learn meaningful features from seen classes and to transfer the knowledge to segment unseen class objects. Then, we alter our MPOT with two comparative matching methods: Bipartite (Kuhn, 1955) and Self-attention algorithm (Vaswani et al., 2017). We find that both methods show inferior performance for unseen classes by margins of above 6% mIoU. The Self-attention method inherently contains learnable projection layers, which yields the model to be overfitted on seen class knowledge. Whereas, MPOT needs no additional trainable parameters, thus our framework can avoid overfitting issues. The bipartite matching performs a one-to-one assignment, where each pixel can be activated for only one prompt, thus it may overlook significant semantic features that can be shared among multiple text prompts. In contrast, MPOT adapts a one-to-many assignment, where each pixel can be partially assigned to multiple prompts, thus it maximizes prompt tuning efficiency by preserving overlapping features among multiple text prompts. The ablation study results demonstrate that our ZegOT achieves the best unseen classes performance by optimally transporting multiple text prompts for dense prediction in the zero-shot segmentation setting.

# 6 CONCLUSION

In this work, we proposed ZegOT, a novel framework for zero-shot semantic segmentation, which thoroughly leverages the aligned vision-language knowledge of a frozen visual-language model by optimizing the multiple text prompts. We also incorporated optimal transport theory into our framework to train multiple text prompts for representing different semantic properties of visual inputs. We demonstrated that our ZegOT outperformed zero-shot segmentation approaches on various benchmark datasets. Our in-depth analyses also confirmed that our ZegOT effectively delivered performance gains on both seen and unseen classes.

**Ethics Statement** We appreciate the valuable research contributions that have supported our study. In our paper, we have taken measures to address potential privacy concerns related to visual results involving individuals. To safeguard privacy, we have anonymized all individuals in our figures by covering their faces.

**Reproducibility** We will release our source code in the final version for reproduction. Additionally, comprehensive details regarding hyperparameters and algorithms for our proposed method can be found in both the main paper and supplementary material.

**Limitation** Our model relying on a small number of learnable parameters has limited room for performance improvement for large-scale datasets. We expect this may be solved in a future study, by introducing an advanced MPOT, which balances the trainable parameters and performance.

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

## A  ALGORITHM OF ZEGOT

---

**Algorithm 1:** Multiple Prompt Optimal Transport Solver with Sinkhorn algorithm

---

**Given:** The feature map size $M = H_L W_L$, the number of prompts $N$, $\boldsymbol{\mu} = \mathbf{1}^M/M$ ,
  $\boldsymbol{\nu} = \mathbf{1}^N/N$, the score map $\boldsymbol{S}$ ;
**Input** :The cost matrix $\boldsymbol{C} = \mathbf{1} - \boldsymbol{S}$, hyper-paramter $\epsilon$, the max iteration $t_{\max}$;
**Initialization:** $\boldsymbol{K} = \exp(-\boldsymbol{C}/\boldsymbol{\epsilon})$, $t \leftarrow 1, \boldsymbol{b}^0 = 1$;
1 **while** $t \leq t_{max}$ and not converge **do**
2     $\boldsymbol{a}^t = \boldsymbol{\mu} \,/\, (\boldsymbol{K}\boldsymbol{b}^{t-1})$;
3     $\boldsymbol{b}^t = \boldsymbol{\nu} \,/\, (\boldsymbol{K}^\top \boldsymbol{a}^t)$;

**Output :**Optimal transport plan $\boldsymbol{T}^* = \mathrm{diag}(\boldsymbol{a})^t \boldsymbol{K} \mathrm{diag}(\boldsymbol{b})^t$ ;

---

## B  ZERO-SHOT SEGMENTATION TRAINING PROCESS.

By following the general transductive zero-shot semantic segmentation (ZS3) setting (Zhou et al., 2022a;d), we divide the entire training procedure into two phases, as described in Algorithm 2.

---

**Algorithm 2:** ZegOT Pseudo-code

---

1 **Input:** ZegOT model $Z_t$ at iteration $t$, The subset of seen classes $\mathcal{C}_S$ and unseen classes $\mathcal{C}_U$, *e.g*,
  $\mathcal{C}_S \cap \mathcal{C}_U = \emptyset$. The training dataset $D = \{(x, \mathbf{Y}^{\mathrm{gt}}) | x \in \mathcal{X}, \mathbf{Y}^{\mathrm{gt}}_{hw} \in \mathcal{C}_S\}$, the training iterations for
  seen class-guided learning $T_g$, and self-training $T_s$;
2 **Phase 1: Seen class-guided learning**
3 **for** $t = 1, 2, \cdots, T_g$ **do**
4     $\mathbf{Y}, \tilde{\mathbf{Y}} \leftarrow$ model prediction from $Z_t(x)$;
5     $\mathcal{L} \leftarrow \mathcal{L}_{\mathrm{seg}}(\mathbf{Y}, \mathbf{Y}^{\mathrm{gt}}) + \mathcal{L}_{\mathrm{seg}}(\tilde{\mathbf{Y}}, \mathbf{Y}^{\mathrm{gt}})$;
6     $Z_{t+1} \leftarrow$ AdamW model parameter update;

7 **Phase 2: Self-training**
8 **for** $t = T_g + 1, T_g + 2, \cdots, T_g + T_s$ **do**
9     $\mathbf{Y}, \tilde{\mathbf{Y}} \leftarrow$ model prediction from $Z_{t-1}(x)$;
10    **if** $\mathbf{Y}^{gt}_{hw} \notin \mathcal{C}_S$ **then**
11     $\mathbf{Y}^{\mathrm{gt}}_{hw} = \underset{\boldsymbol{c} \in \mathcal{C}_U}{\arg\max} \, ((\lambda \mathbf{Y} + (1-\lambda)\tilde{\mathbf{Y}})_{hw} = \boldsymbol{c}|x)$;
12    $\mathbf{Y}, \tilde{\mathbf{Y}} \leftarrow$ model prediction from $Z_t(x)$;
13    $\mathcal{L} \leftarrow \mathcal{L}_{\mathrm{seg}}(\mathbf{Y}, \mathbf{Y}^{\mathrm{gt}}) + \mathcal{L}_{\mathrm{seg}}(\tilde{\mathbf{Y}}, \mathbf{Y}^{\mathrm{gt}})$;
14    $Z_{t+1} \leftarrow$ AdamW model parameter update;

---

**Phase 1. Seen Class-guided Learning**  The model is trained utilizing the ground truth label $\mathbf{Y}^{\mathrm{gt}}$, which only contains pixel-wise labels $\mathbf{Y}^{\mathrm{gt}}_{hw}$ for seen classes $\mathcal{C}_S$, and the rest labels are ignored for calculating losses. $T_g$ is fixed as 20% of total training iterations.

**Phase 2. Self-training**  For the rest of the training iterations, the model self-generates each pixel values of the ground truth label $\mathbf{Y}^{\mathrm{gt}}_{hw}$ which not belongs to $\mathcal{C}_S$, at every training iteration. For large-scale datasets with complex classes, e.g., PASCAL Context and COCO-Stuff164K, we firstly fix $t$ of $Z_{t-1}$ as $T_g$ to ensure stability in the label generation process, and utilize $Z_{T_g-1}$ until $t$ reaches 50% of total training iterations. For the rest of the self-training period, $t$ of $Z_{t-1}$ is updated simultaneously.

## C   DETAILS OF LOSS FUNCTION

As discussed in Section 4.3, we combine three different losses, including the focal loss based on Cross Entropy (CE) loss, and the dice loss, which are given by:

$$\mathcal{L}_{\text{ce}} = -\frac{1}{HW} \sum_{i=1}^{HW} \mathbf{Y}_i^{\text{gt}} \log(\phi(\mathbf{Y}_i))$$

$$+(1 - \mathbf{Y}_i^{\text{gt}}) \log(1 - \phi(\mathbf{Y}_i)), \tag{13}$$

$$\mathcal{L}_{\text{focal}} = -\frac{1}{HW} \sum_{i=1}^{HW} \mathbf{Y}_i^{\text{gt}} (1 - \sigma(\mathbf{Y}_i))^\gamma \log(\sigma(\mathbf{Y}_i))$$

$$+(1 - \mathbf{Y}_i^{\text{gt}}) \sigma(\mathbf{Y}_i)^\gamma \log(1 - \sigma(\mathbf{Y}_i)), \tag{14}$$

$$\mathcal{L}_{\text{dice}} = 1 - \frac{2 \sum_{i=1}^{HW} \mathbf{Y}_i^{\text{gt}} \mathbf{Y}_i}{\sum_{i=1}^{HW} \mathbf{Y}_i^{\text{gt}2} + \sum_{i=1}^{HW} \mathbf{Y}_i^2}, \tag{15}$$

$$\tag{16}$$

where $\mathbf{Y}$ is the model decoder outputs, $\mathbf{Y}^{\text{gt}}$ is the ground truth label, $\sigma(\cdot)$ is Sigmoid operations, $\gamma$ is a hyper-parameter to balance hard and easy samples, which is set to 2. Throughout the entire experiments, $\lambda_{\text{ce}}$, $\lambda_{\text{focal}}$ and $\lambda_{\text{dice}}$ are set to 1, 20, and 1, respectively.

## D   DETAILS OF DATASET

We utilize a total of three datasets, *i.e.,* PASCAL VOC 2012 (Everingham & Winn, 2012), PASCAL Context (Mottaghi et al., 2014), and COCO-Stuff164K (Caesar et al., 2018). We divide seen and unseen classes for each dataset, following the settings of previous methods (Bucher et al., 2019; Xu et al., 2021; Ding et al., 2022; Zhou et al., 2022a;d). PASCAL VOC 2012 consists of 10,582 / 1,449 images with 20 categories, for training / validation. The dataset is divided into 15 seen and 5 unseen classes. PASCAL Context is an extensive dataset of PASCAL VOC 2010 that contains 4,996 / 5,104 images for training / test with 60 categories. The dataset is categorized into 50 seen and 10 unseen classes. COCO-Stuff 164K is a large-scale dataset that consists of 118,287 / 5,000 images for training / validation with 171 classes. The dataset is categorized into 156 seen and 15 unseen classes.

## E   DEFINITION OF HARMONIC MEAN IOU

Following the previous works (Xu et al., 2021; Zhou et al., 2022a;d), we define harmonic mean IoU (hIoU) among the seen (S) and unseen (U) classes as:

$$\text{hIoU} = \frac{2 * \text{mIoU (S)} * \text{mIoU (U)}}{\text{mIoU (S)} + \text{mIoU (U)}} \tag{17}$$

## F   IMPLEMENTATION DETAIL

We further declare the implementation detail for our work. Input image resolution is set as 480×480 for PASCAL Context, and 512×512 for the rest of the datasets. The context length is set to 8 and the weight $\lambda$ for the score map (11) is set to 0.2. We choose the lightest training schedule, which is 20K / 40K / 80K for PASCAL VOC 2012 / PASCAL Context / COCO-Stuff-164K.

## G   NETWORK COMPONENT

**Image Decoder**   Our model output combines the refined score map and the decoder output as in (11). While the score map itself can be the final output, incorporating only the limited number of trainable parameters results in suboptimal performance. To overcome this limitation we introduce the learnable image decoder, which is designed to predict Y in (11). The image decoder is composed of total three

operators, e.g., Feature Pyramid Network (FPN) (Lin et al., 2017), atrous spatial pyramid pooling (ASPP) (Chen et al., 2017) and upsampling operator $\mathcal{U}$, where role of each component is described in Table 5. We further conduct ablation studies on decoder outputs as network logit in Table 6.

Table 5: The detail of image decoder.

| Decoder components | Role |
| --- | --- |
| FPN | Merges intermediate score maps to improve the quality of features. |
| ASPP | Conducts parallel convolutions with different atrous rate to cover a larger receptive field. |
| Upsampling operator | Matches spatial resolution of the output features with that of the label. |

Table 6: Ablation studies of combinations of the network outputs.

| Method | Score map | Decoder output | PASCAL VOC 2012 | | | PASCAL Context | | | COCO-Stuff164K | | |
| --- | --- | --- | --- | --- | --- | --- | --- | --- | --- | --- | --- |
| | | | mIoU (U) | mIoU (S) | hIoU | mIoU (U) | mIoU (S) | hIoU | mIoU (U) | mIoU (S) | hIoU |
| Ours | ✓ | ✓ | **90.9** | **91.9** | **91.4** | **72.5** | **50.5** | **59.5** | **53.9** | **37.0** | **43.8** |
| (a) | ✓ | ✗ | 88.3 | 89.9 | 89.1 | 67.3 | 45.7 | 54.4 | 46.7 | 30.4 | 36.8 |
| (b) | ✗ | ✓ | 89.8 | 91.7 | 90.8 | 69.2 | 47.2 | 56.1 | 51.5 | 34.8 | 41.5 |

**Global Text Alignment (GA)**  The pixel embeddings $\boldsymbol{f}_L$ inherently contain the text-image aligned knowledge of CLIP. However, when pre-training, the global image embedding $\bar{\boldsymbol{f}}_L$ and the text embeddings $\boldsymbol{g}$ comprise the cosine similarity score which is maximized through contrastive learning so that $\bar{\boldsymbol{f}}_L$ contains richer text-pixel aligned information than $\boldsymbol{f}_L$. Thus, we further exploit $\bar{\boldsymbol{f}}_L$ in the GA module to compute the score map $\boldsymbol{S} \in \mathbb{R}^{H_L W_L \times KN}$ by employing an idea of relation descriptor method (Zhou et al., 2022d) in (5).

**Deep Feature Alignment (DFA)**  Since ZegOT adopts FPN, we fully align the extracted intermediate pixel embeddings from the frozen image encoder with the refiend text embedding $\boldsymbol{g}_{GA}$. In other words, we extend the idea of GA to all the intermediate pixel embeddings, *i.e.*, $\{\boldsymbol{S}_i\}_{i=1}^{L}$, where $\boldsymbol{S}_i \in \mathbb{R}^{H_L W_L \times KN}$ is the $i$-th score map. Through the simple arithmetic calculation, the entire hidden image embeddings extracted from the frozen CLIP encoder can be deeply aligned with both the global image and text embeddings without any further learnable parameters.

To demonstrate the effectiveness of the proposed DFA, we conduct in-depth analysis comparing the strength of feature alignment with or without DFA on PASCAL VOC 2012 dataset. Figure 5 shows the bar plot of average text-pixel alignment given a specific class name over the image encoder layer index. To calculate the strength of feature alignment, we extract the score maps $\{\boldsymbol{S}_i|_{i=1}^{L}\}$ for a certain class name (*e.g.*, dog) from the trained model and calculate the average $\boldsymbol{S}_i$ along entire pixels for each image encoder layer. We find that the strength of feature alignment with DFA is much higher compared to that computed through the model without DFA in almost layers. In particular, the average value of text-pixel alignment significantly increases at the earlier layers and the final layer. The result confirms that our model with DFA effectively boosts the text-pixel alignment given intermediate pixel embeddings from the frozen CLIP encoder.

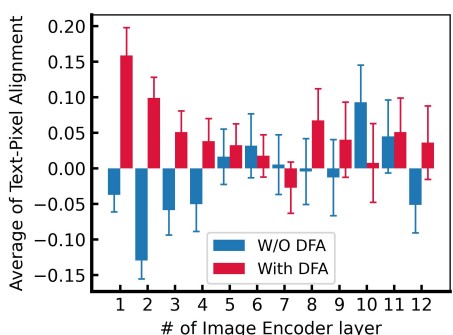

Figure 5: Bar plots of the average text-pixel alignment versus the image encoder layer index given a specific class name with or without DFA.

## H COMPONENT ANALYSIS

To study the effects of hyper-parameters of network components on the zero-shot segmentation performance, we conduct the component analysis in Table 7 with varying hyper-parameters including:

MPOT layers ($L$), number of text prompts ($N$), context length ($C$), and balance factor ($\lambda$). Firstly, we investigate the effect of incorporating layers $L$ for the MPOT solver, *i.e.,* layers of the frozen image encoder that inserted to MPOT. Although inserting multiple layers to MPOT boosts performance, it also causes performance trade-off between seen and unseen classes segmentation. We find that introducing the MPOT with 8 to 12-th layers of the image encoder achieves the best performance, which becomes the default setting of ZegOT for the entire experiments. We further explore the effect of the depth of MPOT solver, *i.e.,* ranges of the frozen image encoder layers that inserted to MPOT. Further, we observe the segmentation performance by varying the total number $N$ of the multiple text prompts. We empirically find that ZegOT performs the best when $N \geq 4$, but the hIoU performance drops when $N$ is increased over certain threshold, which implies that fewer text prompts are insufficient to learn comprehensive semantic features, whereas too many text prompts complicate the optimal transport matching process. Moreover, we empirically find that the large context length ($C$) is limited to take significant effect on segmentation performance, *e.g,* when $C = 16$ the performance on seen classes is the best, but the performance on unseen classes drops. Since we consider the hIoU as a major metric, we adopt $C = 8$ as the default setting. Lastly, we empirically search the balance factor $\lambda$ for (11) to control weight between the image decoder outputs and the score maps. we observe that our method is robust for various parameter values, and we observed that setting lambda to 0.8 yields the best performance for our method.

Table 7: Component analysis of hyper-parameters. # denotes the hyper-parameters of configurations. Checkmark ✓ indicates the default configuration of ZegOT.

| Components | # | PASCAL VOC 2012 | | | | Components | # | PASCAL VOC 2012 | | | |
|---|---|---|---|---|---|---|---|---|---|---|---|
| | | mIoU (U) | mIoU (S) | hIoU | Ours | | | mIoU (U) | mIoU (S) | hIoU | Ours |
| MPOT layers (L) | 12 | 91.8 | 87.3 | 89.5 | | Balance factor ($\lambda$) | 1 | - | - | 88.9 | |
| | 10-12 | 91.4 | 87.3 | 89.3 | | | 0.9 | - | - | 91.3 | |
| | 8-12 | 90.9 | **91.9** | **91.4** | ✓ | | 0.8 | - | - | **91.4** | ✓ |
| | 4-12 | 91.6 | 90.2 | 90.9 | | | 0.7 | - | - | 91.3 | |
| | 1-12 | **92.1** | 86.5 | 89.3 | | | 0.6 | - | - | 91.1 | |
| Text prompts (N) | 2 | 82.9 | 91.6 | 87.0 | | Context lengths (C) | 8 | **90.9** | 91.9 | **91.4** | ✓ |
| | 4 | **90.9** | **91.9** | **91.4** | ✓ | | 16 | 89.6 | **92.2** | 90.9 | |
| | 6 | 89.6 | 91.8 | 90.7 | | | 32 | 90.3 | 91.9 | 91.1 | |

## I   ABLATION STUDIES ON PIXEL-TEXT PROMPTS ALIGNMENT.

We conduct ablation experiments to evaluate the alignment of pixel-text prompts as shown in Table 8. (a) To investigate the proposed framework's synergistic effect, we replace the learnable text prompt with hand-crafted prompts. Our model, equipped with all proposed components, achieves superior performance with margins of 10.6 % hIoU and 7.9 % hIoU for PASCAL VOC 2012 and PASCAL Context dataset, respectively. (b) We further ablate the MPOT module and the results show no significant difference to those of (a), which implies that our MPOT module can boost the segmentation performance when incorporated with learnable text prompts as shown in Table 8-Ours. (c) Lastly, as discussed in the Section G, we examine the impact of the global text-alignment (GA) layer. Interestingly, when using hand-crafted prompts, removing the GA layer leads to a significant decrease in performance with margins of 50.3 % hIoU, and 31.4 % hIoU for each PASCAL VOC 2012, and PASCAL Context dataset, respectively. These findings validate the effectiveness of the proposed framework, which incorporates learnable text prompts with the GA layer and the MPOT solver, by optimally aligning text-pixel embeddings to achieve the best performance in the zero-shot segmentation task.

## J   ANALYSIS ON THE EFFICIENCY OF MPOT

To analyze the efficiency of MPOT, we measured the total memory latency and training complexity (GFLOPS) with and without MPOT. As described in Table 9, our method with MPOT yields outperforming results by sacrificing only a small amount of computational cost. Additionally, we measured the actual runtime for inference on an NVIDIA GeForce RTX 3090 using the PASCAL VOC dataset. Despite the slight increase in runtime latency, our method yields superior performance compared to the method without MPOT, as shown in Table 9.

Table 8: Ablation studies of the network components

| Method | MPOT | GA | Text prompt | PASCAL VOC 2012 | | | PASCAL Context | | |
|---|---|---|---|---|---|---|---|---|---|
| | | | | mIoU (U) | mIoU (S) | hIoU | mIoU (U) | mIoU (S) | hIoU |
| Ours | ✓ | ✓ | Learnable | **90.9** | **91.9** | **91.4** | **72.5** | **50.5** | **59.5** |
| (a) | ✓ | ✓ | Hand-crafted | 78.1 | 83.7 | 80.8 | 54.9 | 33.5 | 41.6 |
| (b) | ✗ | ✓ | Hand-crafted | 77.8 | 83.7 | 85.4 | 54.7 | 33.4 | 41.4 |
| (c) | ✗ | ✗ | Hand-crafted | 33.9 | 36.3 | 35.1 | 15.2 | 7.5 | 10.0 |

Furthermore, we compared the previous approaches in terms of actual runtime for inference. Despite the slight increase in runtime latency compared to ZegCLIP, our method is superior compared to the method including Zsseg, and ZegFormer as indicated in Table 10.

Table 9: Comparison of memory cost and inference time.

| Method | Training memory latency (M) | Training Complexity (GFLOPS) | Actual runtime for inference (images per second) |
|---|---|---|---|
| Without MPOT | 13028 | 147.1 | 9.6 |
| With MPOT | 13914 | 148.9 | 8.9 |
| (+computational cost) | (+6.8%) | (+1.2%) | - |

Table 10: Comparison of inference time to previous approaches.

| Method | Actual runtime for inference (images per second) |
|---|---|
| Zsseg | 4.24 |
| ZegFormer | 6.86 |
| ZegCLIP | 10.8 |
| Ours | 8.9 |

## K  ANALYSIS ON NORMALIZATION METHOD OF MPOT

The purpose of utilizing Sigmoid normalization is to ensure stable calculations in conjunction with the geometric mean. To further support our claim, we provide additional comparative results with the following cases: 1) Inferencing the pre-trained model performance using Min-max normalization to scale the values within the range [0,1], and 2) Training the model using Min-max normalization. The results in Table 11 validate the robustness and effectiveness of the Sigmoid normalization used within our framework.

Table 11: Comparison of Normalization method of MPOT

| Normalization method | Pascal VOC 2012 | | |
|---|---|---|---|
| | mIoU (U) | mIoU (S) | hIoU |
| Sigmoid (Ours) | **90.9** | **91.9** | **91.4** |
| 1) Min-max (inference-only) | 90.1 | 91.9 | 91.0 |
| 2) Min-max (train & inference) | 37.4 | 91.7 | 53.1 |

## L   FURTHER EXPERIMENTS ON INDUCTIVE SETTING

In general, zero-shot semantic segmentation performance in the inductive setting is quite limited to beat the supervised methods. In this paper, we focus on improving zero-shot segmentation performance which can be comparable to that of the supervised learning methods, thus we conduct experiments under the transductive setting. Nevertheless, we have implemented the proposed framework in the inductive setting and proved the comparable results to SOTA methods as in the Table 12.

Table 12: Performance comparison for the inductive setting.

| Method | Pascal VOC 2012 | | |
|---|---|---|---|
| | mIoU (U) | mIoU (S) | hIoU |
| ZegFormer | 63.6 | 86.4 | 73.3 |
| Zsseg | 72.5 | 83.5 | 77.6 |
| ZegCLIP | 77.8 | 91.9 | 84.3 |
| Ours | 71.6 | 86.5 | 78.4 |

## M   EXPERIMENT FOR INCORPORATING MPOP AS A PLUG-IN MODULE

We conducted additional experiments to analyze the effectiveness of MPOT as a plug-in module by integrating it into ZegCLIP. Due to the intrinsic nature of the framework, which includes the network architecture, we incorporated MPOT as an additional pipeline for computing loss. Adhering to the original scheme of ZegCLIP, we used multiple hand-crafted prompts, maintaining the same number of prompts as 4. The comparison results are presented in Table 13, demonstrating that our proposed MPOT serves as a plug-in module, further providing an extra gain for the SOTA model.

Table 13: Performance of MPOT as a plug-in module for ZegCLIP. † indicates the reproduced result for ZegCLIP.

| Method | Multiple Text Prompts | MPOT | PASCAL VOC 2012 | | |
|---|---|---|---|---|---|
| | | | mIoU (U) | mIoU (S) | hIoU |
| ZegCLIP | ✗ | ✗ | 89.9 | 92.3 | 91.1 |
| ZegCLIP† | ✗ | ✗ | 89.7 | 92.2 | 90.9 |
| (a) | ✓ | ✗ | 92.3 | **93.5** | **92.9** |
| (b) | ✓ | ✓ | **93.0** | 92.8 | **92.9** |

## N   ADDITIONAL VISUAL RESULTS

We provide additional visual results. Figure 6 shows further visual results of MPOT module contribution to segmentation. Figure 7 show qualitative zero-shot segmentation performance of our ZegOT for PASCAL Context. For PASCAL Context dataset, we reproduce the segmentation results using a publically available model weight (Zhou et al., 2022a). Our ZegOT demonstrates superior performance in precisely sectioning boundaries of unseen objects, unlike previous SOTA method which fail to classify the category (see the red arrows). Figure 8 show further qualitative zero-shot segmentation performance for COCO-Stuff164K dataset. We reproduce the segmentation results using publically available weights (Zhou et al., 2022a;d).

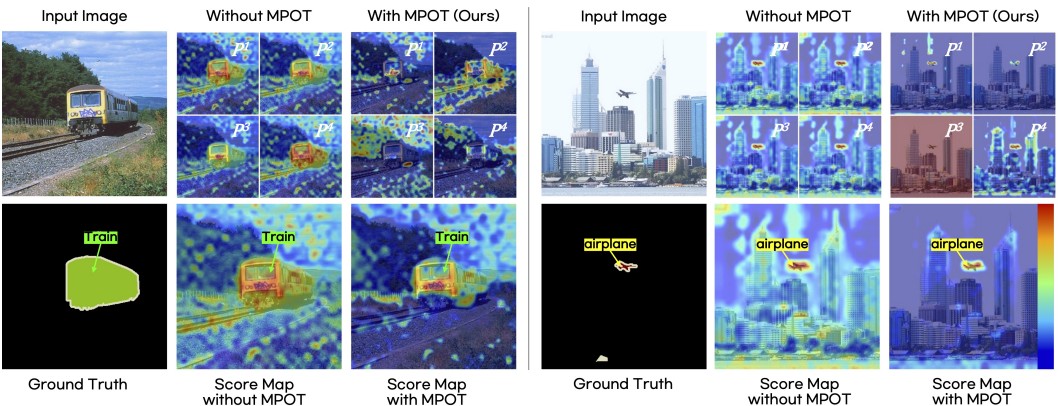

Figure 6: Further visual results of text-pixel alignment of MPOT module for both unseen and seen classes. The green tag indicates unseen classes, while the yellow tag indicates seen classes. Without proposed MPOT, all the learned multiple prompts $P^i$ are cohered and their related score maps resemble each others. On the other hand, with our MPOT, each $P^i$ is diversely projected and their related score map focuses on different semantic attributes. As results, the final score map with MPOT effectively focuses on the ground truth regions compared to that of without MPOT.

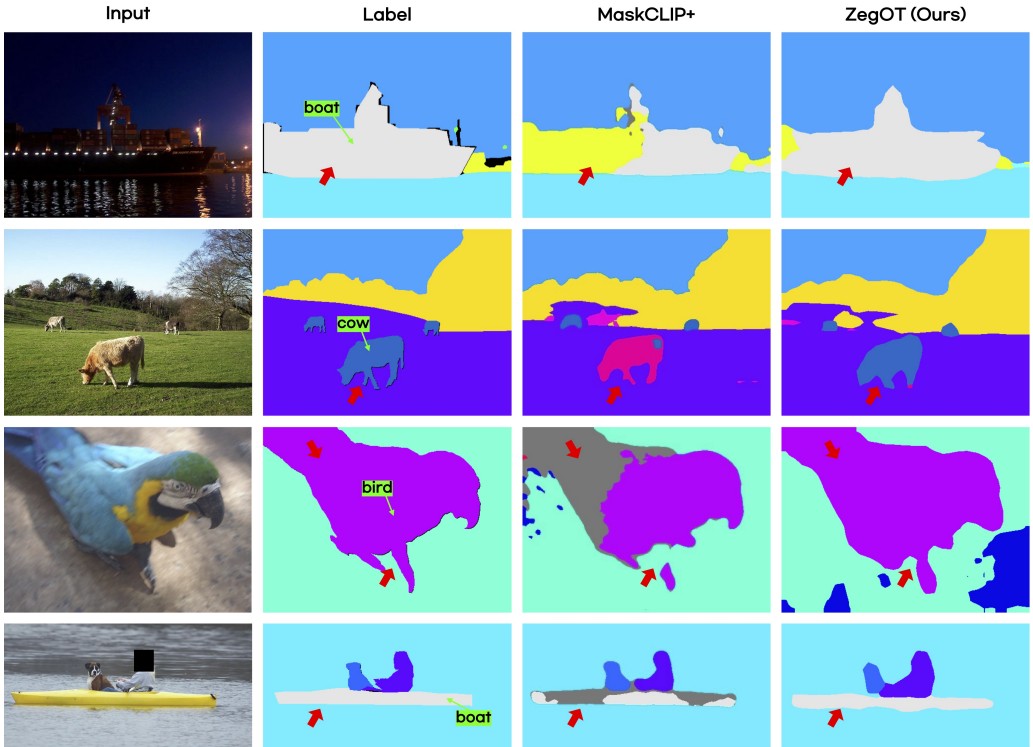

Figure 7: Qualitative zero-shot segmentation results of PASCAL Context dataset. The green tag indicates unseen classes. Red arrows indicate superior segmentation results of our ZegOT by precisely sectioning boundaries of unseen objects compared to the previous SOTA method.

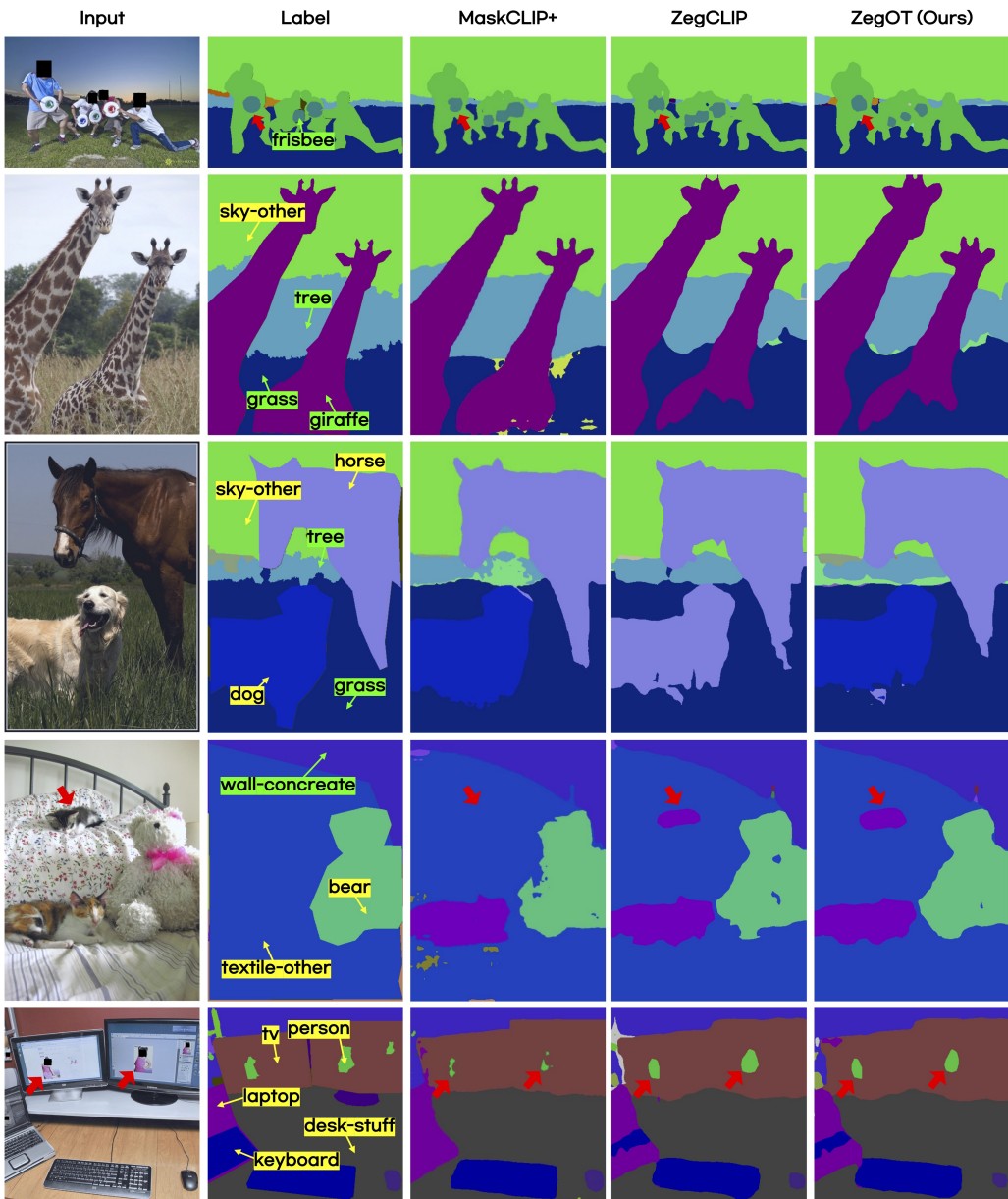

Figure 8: Qualitative zero-shot segmentation results of COCO-Stuff164K dataset. The `yellow` tag indicates seen classes, while the `green` tag indicates unseen classes. Our ZegOT precisely sections boundaries of unseen objects compared to the previous SOTA method (see red arrows in 1st row). Moreover, our ZegOT effectively segments tiny cat category that does not even belong to the ground truth (see red arrows in 4th row), and properly segment the person category within the small-sized portraits (see red arrows in 5th row).

