# OpenReview forum: "ZegOT: Zero-shot Segmentation Through Optimal Transport of Pixels to Text Prompts"
_ICLR.cc/2024/Conference — Submitted to ICLR 2024_

### Official Review · Reviewer_41mH · 2023-10-30

**Soundness:** 2 fair
**Presentation:** 2 fair
**Contribution:** 2 fair
**Rating:** 3
**Confidence:** 4

**Summary:**

The paper proposes a novel method called ZegOT for zero-shot semantic segmentation, which transfers text-image aligned knowledge to pixel-level classification without the need for additional image encoders or retraining the CLIP module. ZegOT utilizes a Multiple Prompt Optimal Transport Solver (MPOT) to learn an optimal mapping between multiple text prompts and pixel embeddings of the frozen image encoder layers. This allows each text prompt to focus on distinct visual semantic attributes and diversify the learned knowledge to handle previously unseen categories effectively.

**Strengths:**

+ The experimental results lead the existing state-of-the-art (SOTA)  under the same settings on some datasets.


+ The author modeled the optimal transport problem into the segmentation task of open vocabulary, providing a new approach, and this module can alleviate the problem of overfitting to the seed class.

**Weaknesses:**

-	The technical insight may not be enough The Deep Feature Alignment module proposed by the author is equivalent to extending the Relationship Descriptor based on Zegclip to multi level features, with the core still being the Relationship Descriptor. In addition, similar to CoOp's text prompt learning, would it be better to directly apply it to previous methods such as ZegClip?

-	The experiment setting is not clear to me. For example, If the proposed method can effectively solve the problem of overfitting the network parameters to the seed class distribution after training, why not conduct a set of experiments under the setting of Inductive (unseen class names and images are not accessible during training.). In Table 2, the mIoU (U) and miou (S) of ZegCLIP are 87.3 and 92.3 respectively, but hiou should not be 91.1 and should be 89.7. The setting used in the experiment in table2 is Conductive. But does Tabel3 seem to be using Inductive settings? No explanation was provided, and would using Transductive settings be better than the previous method. In addition, the author claims to have obtained a sota, but it is 2.2 hiou lower on COCO-Stuff164K. For table4, I would like to know how much improvement can be achieved by using only MPOT compared to baseline.


-	Formula 17 is written incorrectly.

**Questions:**

seeing Weaknesses

---

> ### Author Response · Authors · 2023-11-19
> **Reply to Reviewer 41mH**
>
> >**W1-1**. The technical insight may not be enough. The Deep Feature Alignment module proposed by the author is equivalent to extending the Relationship Descriptor based on Zegclip to multi level features, with the core still being the Relationship Descriptor.
>
> -> As the reviewer correctly pointed out, DLFA is an extension of the Relationship Descriptor in ZegCLIP. However, in this work, we introduce the concept of multiple-level applications of the Relationship Descriptor for the first time. Additionally, we would like to kindly emphasize that the novelty of ZegOT extends beyond DLFA, encompassing MPOT and thorough technical analysis. We encourage you to explore the various aspects of our proposed methodology beyond DLFA.
>
> >**W1-2**. In addition, similar to CoOp's text prompt learning, would it be better to directly apply it to previous methods such as ZegClip?
>
> -> Thanks for your concrete suggestion. Per your suggestion, we conducted the experiment to incorporate MPOT into ZegCLIP. Due to the instinctual nature of the framework including network architecture, we integrate MPOT into the last layer of decoder in ZegCLIP.  To follow the original scheme of ZegCLIP, we use the multiple hand-crafted prompts and use the same number of multiple prompts, as 4. The comparison results are presented in the following table, and add the related contents in the revised manuscript.
>
> ***Table.*** Performance of ZegCLIP + MPOT on PASCAL datasets.
> | Methods         | Multiple Prompts (N=4) | IoU(U) | IoU(U) | IoU(S) | hIoU |
> |-----------------|:------:|:------:|:----:|:----:|:----:|
> | ZegCLIP           |  X |  X  |89.9| 92.3 | 91.1 |
> | ZegCLIP(reproduced) |  X |  X | 89.7 | 92.2 | 90.9 |
> | ZegCLIP (a)        |  O  |  X  | 92.3 | 93.5 | 92.9 |
> | ZegCLIP (b) |  O |  O  | 93.0 | 92.8 | 92.9 |
>
> >**W2-1**. The experiment setting is not clear to me. For example, If the proposed method can effectively solve the problem of overfitting the network parameters to the seed class distribution after training, why not conduct a set of experiments under the setting of Inductive (unseen class names and images are not accessible during training.).
>
> ->  In general, zero-shot semantic segmentation performance in the inductive setting is quite limited to beat the supervised methods and be directly utilized in practice. In this paper, we focus on improving zero-shot segmentation performance which can be comparable to that of the supervised learning methods, thus we conduct experiments under the transductive setting. By efficiently utilizing only class names for unseen classes, our methodology under the transductive setting boosts overall segmentation performance. Nevertheless, per the reviewer’s suggestion, we have implemented the proposed framework in the inductive setting and proved the comparable results to SOTA methods as in the following table.
>
> ***Table 1.*** Performance comparison on PASCAL VOC 2012 dataset for the Inductive setting.
> | Methods   | IoU(U) | IoU(S) | hIoU |
> |-----------|:------:|:------:|:----:|
> | ZegFormer |  63.6  |  86.4  | 73.3 |
> | Zsseg     |  72.5  |  83.5  | 77.6 |
> | ZegCLIP   |  77.8  |  91.9  | 84.3 |
> | Ours      |  71.6  |  86.5  | 78.4 |
> >**W2-2**.In Table 2, the mIoU (U) and mIoU (S) of ZegCLIP are 87.3 and 92.3 respectively, but hIoU should not be 91.1 and should be 89.7.
>
> -> Thanks for your careful comments. We have updated the results of ZegCLIP in the revised paper.
>
> >**W2-3**. The setting used in the experiment in table2 is Conductive. But does Table 3 seem to be using Inductive settings? No explanation was provided, and would using Transductive settings be better than the previous method.
>
> -> Thanks for your careful observation and valuable recommendation. As the reviewer pointed out, we trained our model using transductive settings on the source dataset and solely performed inference on the target test set. To ensure a fair comparison with other methods, we employ the weight of ZegCLIP, which is trained using transductive settings.
>
> >**W2-4**. In addition, the author claims to have obtained a sota, but it is 2.2 hIoU lower on COCO-Stuff164K.
>
> -> Thanks for your valuable comments. Per your suggestion, we have narrowed the claim down and revised it.
>
> >**W2-5**. For table4, I would like to know how much improvement can be achieved by using only MPOT compared to baseline.
>
> ->The reviewer is kindly reminded that the results you referred to are already indicated in Table 4 (b) and marked with an 'X'. These results confirm that our proposed MPOT significantly enhances performance, demonstrating a substantial margin of improvement with 2.9 and 2.4 mIoU on the PASCAL VOC and Pascal Context datasets, respectively.
>
> >**W3**.Formula 17 is written incorrectly.
>
> -> Thanks for your detailed comments. We have corrected this formulation in the revised paper.

---

> > ### Comment · Reviewer_41mH · 2023-11-21
> > **Response to authors**
> >
> > I appreciate the author's efforts in responding and conducting additional experiments. However, after a thorough review of their responses and the concerns raised by other reviewers, I still believe that this paper does not meet the high threshold of significant contribution required for acceptance at ICLR, even with the new experimental data.
> >
> >
> > In Table 12, the results of the proposed method in the inductive setting significantly lag behind the current state-of-the-art. If the method is effective for zero-shot semantic segmentation, it should not result in a decrease in accuracy.
> >
> >
> > In Table 13, the use of MPOT shows a negligible improvement, equivalent to the hIoU metric without MPOT, and is comparable to ZegCLIP with multiple text prompts.
> >
> >
> > Therefore, I hold my initial assessment of the manuscript.

---

> ### Author Response · Authors · 2023-11-21
> **Dear Reviewer 41mH**
>
> As the deadline for the Reviewer-Author discussion phase is fast approaching (there is only a day left), we respectfully ask whether we have addressed your questions and concerns adequately.

---

### Official Review · Reviewer_Dhw2 · 2023-11-01

**Soundness:** 3 good
**Presentation:** 3 good
**Contribution:** 3 good
**Rating:** 6
**Confidence:** 3

**Summary:**

This paper utilizes the large-scale CLIP model to solve the zero-shot semantic segmentation task. In this paper, the authors have proposed a novel Multiple Prompt Optimal Transport Solver (MPOT) module, which is designed  to learn an optimal mapping between multiple
text prompts and pixel embeddings of the frozen image encoder layers.

**Strengths:**

1. The proposed method solves zero-shot segmentation from a new perspective. They propose optimal transport to enable the alignment between text and pixel space.

**Weaknesses:**

1. The authors include ZegCLIP to " trainable image encoder-based approaches". However, they fix the image encoder and train a new decoder. Such a statement is not accurate.
2. In related work, the authors do not introduce open vocabulary semantic segmentation, which is highly related to this topic in this paper.
3. The performance of COCO-Stuff does not outperform ZegCLIP.  It seems that the proposed method is more useful on simple images such as PASCAL VOC.
4. Lack of inference speed comparison with previous methods. As the authors propose several blocks into the ZS3 framework,  it is quite essential to consider the inference speed.
5. The optimal transport plan is just like a spatial attention map for each class and each prompt, I guess adding a self-attention layer or learnable spatial attention maps is also effective.

**Questions:**

It seems that the proposed Optimal Transport-based method can be plugged into other methods, such as ZegCLIP. Is it possible to use Optimal Transport in ZegCLIP?

---

> ### Author Response · Authors · 2023-11-19
> **Reply to Reviewer Dhw2**
>
> >**W1**. The authors include ZegCLIP to "trainable image encoder-based approaches". However, they fix the image encoder and train a new decoder. Such a statement is not accurate.
>
> -> Thanks for your careful comment. This was because ZegCLIP employs a vision prompt tuning approach for tuning the image encoder, which we consider as a trainable encoder.  That being said, we agree with the reviewer, so we now categorize ZegCLIP as a frozen image encoder-based approach.
>
> >**W2**. In related work, the authors do not introduce open vocabulary semantic segmentation, which is highly related to this topic in this paper.
>
> -> Thanks for your concrete and helpful comment. We have expanded the scope of zero-shot semantic segmentation to include open-vocabulary semantic segmentation topic in the related work section 2.2 in the revised paper.
>
> >**W3**.The performance of COCO-Stuff does not outperform ZegCLIP. It seems that the proposed method is more useful on simple images such as PASCAL VOC.
>
> -> As the reviewer pointed out, we admit that relying on a few learnable text prompts and lightweight decoder with limited trainable parameters limits the room for performance improvement on a large-scale dataset, compared to the retraining/fine-tuning of the image backbone. We expect this may be solved in a future study, by introducing an advanced MPOT, which balances the trainable parameters and performance for large-scale datasets. We have updated this limitation in the revised paper.
>
> >**W4**. Lack of inference speed comparison with previous methods. As the authors propose several blocks into the ZS3 framework, it is quite essential to consider the inference speed.
>
> -> Per the reviewer’s valuable comment, we have measured actual runtime for inference on an NVIDIA GeForce RTX 3090 using the PASCAL VOC dataset. Despite the slight  increase in runtime latency compared to ZegCLIP, our method is superior compared to the method including Zsseg, and ZegFormer as shown in the following table.
>
> ***Table 4.*** Comparison inference speed with other methods on the same device, a single GeForce RTX 3090 device.
> | Methods         | inference speed (image per seconds)|
> |-----------------|:------:|
> | Zsseg           |  4.24 |
> | ZegFormer |  6.86 |
> | ZegCLIP        |  10.8  |
> | Ours |  8.9 |
>
> >**W5**. The optimal transport plan is just like a spatial attention map for each class and each prompt, I guess adding a self-attention layer or learnable spatial attention maps is also effective.
>
> -> The reviewer is kindly reminded that a comparison with the results of the self-attention layer is already presented in Table 4 (b) of the main paper, as indicated  'Self-attention'. Additionally, we compare the results with other matching methods using a one-to-one mapping, referred to as the Bipartite matching method, within the same table. As you correctly commented, both the Self-attention and the Bipartite matching methods yield meaningful results; however, our proposed ZegOT outperforms these methods on both the Pascal VOC and Pascal Context datasets.
>
> >**Q1**. It seems that the proposed Optimal Transport-based method can be plugged into other methods, such as ZegCLIP. Is it possible to use Optimal Transport in ZegCLIP?
>
> -> Thanks for your concrete suggestion. Per your suggestion, we conducted the experiment to incorporate MPOT into ZegCLIP. Due to the instinctual nature of the framework including network architecture, we integrate MPOT as an additional pipeline for computing loss. To follow the original scheme of ZegCLIP, we use the multiple hand-crafted prompts and use the same number of multiple prompts, as 4. The comparison results are presented in the following table, and we demonstrate that our proposed MPOT acts as a plug-in module for further boosting the performance of the SOTA model. We have added the related contents in the revised manuscript.
>
> ***Table 5.*** Performance of ZegCLIP + MPOT on PASCAL datasets.
> | Methods         | Multiple Prompts (N=4) | IoU(U) | IoU(U) | IoU(S) | hIoU |
> |-----------------|:------:|:------:|:----:|:----:|:----:|
> | ZegCLIP           |  X |  X  |89.9| 92.3 | 91.1 |
> | ZegCLIP(reproduced) |  X |  X | 89.7 | 92.2 | 90.9 |
> | ZegCLIP (a)        |  O  |  X  | 92.3 | 93.5 | 92.9 |
> | ZegCLIP (b) |  O |  O  | 93.0 | 92.8 | 92.9 |

---

> > ### Comment · Reviewer_Dhw2 · 2023-11-22
> >
> > Thanks for the authors' detailed replies. Although the idea is good, the performance (especially for COCO-Stuff) is still the biggest problem of this paper, attracting the attention of another reviewer.  Therefore, I keep my rating score.

---

### Official Review · Reviewer_kYrk · 2023-11-04

**Soundness:** 3 good
**Presentation:** 3 good
**Contribution:** 3 good
**Rating:** 6
**Confidence:** 4

**Summary:**

The paper proposes a framework that utilizes a frozen vision-language model (CLIP) for zero-shot semantic segmentation. The proposed method learns a set of text prompts that align with pixel embedding at different scales. Since the learn text prompts have coherence and usually have similar score maps with image features, they propose to refine the score map with optimal transport to make them more diverse. The method shows great performance on ZS3 under Pascal VOC, and Pascal Context.

**Strengths:**

- The idea is interesting. They show that learning multiple prompts leads to similar score map for each prompt. Then they propose a method to make different prompts to have different score maps, which improves the diversity of the score maps. Optimal transport fits this purpose. An adequate comparison with other alignment methods such as bipartite matching and self-attention has been shown in the ablation study.

**Weaknesses:**

- What is the reason behind the similarity of the multiple learned prompts?
- The performance of ZegCLIP is different from that in the original paper.
- It seems the method is trained in a transductive setting; What is the performance in an inductive setting?
- How does this work under different backbones? Previous works sometimes use Resnet for ZS3. Please consider a comparison.
- The number of classes seems small. What are the results on datasets with larger numbers of classes, such as Pascal Context 459 and ADE-847?

**Questions:**

Please see weakness.

---

> ### Author Response · Authors · 2023-11-19
> **Reply to Reviewer kYrk**
>
> >**W1**. What is the reason behind the similarity of the multiple learned prompts?
>
> -> Thanks for your valuable comments. If we do not use our proposed optimal transport, the approach of using multiple learnable text prompts becomes aligning the average features of text prompts with the visual features, yielding all the text prompts cohered. Unfortunately, this cohered text prompts are not well-aligned with diverse features of semantic visual embeddings. Similar negative observations were made in the PLOT, where they conducted image classification with multiple prompts [1]. This is our main motivation of using optimal transport for multi-prompt alignment.
>
> [1] Chen et.al. PLOT: Prompt Learning with Optimal Transport for Vision-Language Models, ICLR2023
>
> >**W2**. The performance of ZegCLIP is different from that in the original paper.
>
> -> Thanks for your careful observation. We have updated the latest results of ZegCLIP in the revised paper.
>
>
> >**W3**. It seems the method is trained in a transductive setting; What is the performance in an inductive setting?
>
> -> In general, zero-shot semantic segmentation performance in the inductive setting is quite limited to beat the supervised methods, so that it cannot be directly utilized in practice. Accordingly, our methodology has focused on improving zero-shot segmentation performance, which can be comparable to that of the supervised learning methods. Consequently, we conduct experiments under the transductive setting, by efficiently utilizing only class names for unseen classes to boost overall segmentation performance. Nevertheless, per the reviewer’s suggestion, we have implemented the proposed framework in the inductive setting and proved the comparable results to SOTA methods as in the following table.
>
>
>
> ***Table 1.*** Performance comparison on PASCAL VOC 2012 dataset for the Inductive setting.
> | Methods   | IoU(U) | IoU(S) | hIoU |
> |-----------|:------:|:------:|:----:|
> | ZegFormer |  63.6  |  86.4  | 73.3 |
> | Zsseg     |  72.5  |  83.5  | 77.6 |
> | ZegCLIP   |  77.8  |  91.9  | 84.3 |
> | Ours      |  71.6  |  86.5  | 78.4 |
>
> >**W4**. How does this work under different backbones? Previous works sometimes use Resnet for ZS3. Please consider a comparison.
>
> -> Thanks for the constructive comment. We investigated different backbones used in various baseline models for ZS3  as shown in Table 2 below, and the ResNet backbones are mostly utilized as feature backbone. That being said, we don’t need the feature backbone, since our architecture is VLM and decoder-only architecture. Accordingly, we believe that  the suggested experiments with different backbone are not necessary to verifying the performance of our algorithm.
>
>
> ***Table 2.*** Different backbones for ZS3.
> | Model     | VLM(CLIP) | Feautre Backbone | Decoder |
> |-----------|:------:|:------:|:----:|
> | zsseg |  ViT |  ResNet  | - |
> | ZegFormer     |  ViT  |  ResNet  | FPN |
> | FreeSeg   |  ViT  |  ResNet  | - |
> | MVP-SEG      |  ViT  |  DeepLab  | - |
> | ZegCLIP      |  ViT  |  -  | Light-weight ViT  |
> | Ours      |  ViT  |  -  | FPN / ASPP |
>
> >**W5**. The number of classes seems small. What are the results on datasets with larger numbers of classes, such as Pascal Context 459 and ADE-847?
>
> -> Thanks for the great comment. To demonstrate our proposed model’s generalization performance on large-scale datasets with larger numbers of classes, we conduct open vocabulary segmentation by training the model with full COCO-171. We are running the experiments now, however, due to the limited computational resources, the results are not prepared. We will try to update the results before the end of the rebuttal period.

---

### Author Response · Authors · 2023-11-19
**General Reply to All Reviewers**

We appreciate the constructive comments from the reviewers that helped us in improving our paper. Most of the reviewers commented that this paper is sufficiently novel, and the core idea is interesting to solve zero-shot segmentation from a new perspective, but point out the comparable performance to the previous SOTA model as a weakness. To address the comments, we have conducted the following major experiments.

1.	We conducted the experiment to incorporate our proposed MPOT into ZegCLIP (the previous SOTA model).  We demonstrate that our proposed MPOT acts as a plug-in module for further boosting the performance of the SOTA model.

2.	We implemented the proposed framework in the inductive setting and updated it in the revised paper. We would like to highlight that our methodology has focused on the transductive setting, which can be comparable to that of the supervised learning methods.

3.	We narrow the claim down to achieving SOTA for all the datasets, since our method beats the second-best performance for the large-scale dataset.

Modified contents have been highlighted blue in the revised paper.

**Abstract and Introduction**. We narrowed the claim down to achieving SOTA on all the dataset.

**Section 2.2**. We added more description of open-vocabulary segmentation.

**Table 2**. We updated the results of ZegCLIP.

**Limitation**. We updated the limitation of our model for large-scale datasets.

**Appendix E**. We revised the formulation of harmonic mean for evaluation metric.

**Appendix J**.  We added a comparison on actual inference speed with previous approaches.

**Appendix L**.  We added a result for the inductive setting on the PASCAL VOC12 dataset.

**Appendix M**.  We added results of incorporating MPOT into previous SOTA, i.e., ZegCLIP.

---

### Meta-Review · Area_Chair_Mmd6 · 2023-12-07

**Metareview:**

The paper proposes a zero-shot segmentation method using a frozen vision-language model (CLIP). The method learns a set of text prompts that align with pixel embedding at different scales allowing each text prompt to focus on distinct visual semantic attributes and diversify the learned knowledge to handle previously unseen categories more effectively.

The paper received two weak accepts and a reject. While all three reviewers acknowledged the novelty of the proposed method, the reviewers pointed out the limitation in its scalability. On one hand, the reviewers kYrk and Dhw2 both raised concerns about its effectiveness on larger scale benchmarks with more classes. On the other hand, two reviewers kYrk and 41mH ask about the effectiveness in an inductive setting. Although the authors argue that the main focus of the paper is on the transductive setting, they try to use the full open-vocab setting, which is necessarily an inductive setting, to rebut the limited scalability of the proposed method. Furthermore, the authors partly acknowledged the limitation in the scalability and therefore, this issue remains unaddressed.

Overall, the AC believes the flaws outweigh the benefits and recommends the rejection of the paper.

**Justification For Why Not Higher Score:**

The reviewers raised a common issue, which was not fully addressed during the rebuttal.

**Justification For Why Not Lower Score:**

N/A

---

### Decision · Program_Chairs · 2024-01-16

Reject